# Measuring the Residual Levels of Fenpyroximate and Its Z-Isomer in Citrus Using Ultra-High-Performance Liquid Chromatography–Tandem Mass Spectrometry and Assessing the Related Dietary Intake Risks

**DOI:** 10.3390/molecules28207123

**Published:** 2023-10-17

**Authors:** Ruiqing Sun, Junli Cao, Jindong Li, Yanli Qi, Shu Qin

**Affiliations:** Shanxi Center for Testing of Functional Agro-Products, Longcheng Campus, Shanxi Agricultural University, No. 79, Longcheng Street, Taiyuan 030031, China; cjlagriculture@163.com (J.C.); ljdchemistry@163.com (J.L.); qiyanli0412@126.com (Y.Q.)

**Keywords:** fenpyroximate, citrus, pesticide residues, dietary intake risk

## Abstract

Fenpyroximate is an efficient, broad-spectrum phenoxypyrazole acaricide which is used for controlling various mites. In this study, we measured the levels of terminal fenpyroximate residues in citrus fruits, and estimated the dietary intake risks posed by fenpyroximate. To this end, a QuEChERS analytical method was used in combination with ultra-high-performance liquid chromatography–tandem mass spectrometry (UHPLC-MS/MS) to determine the residual levels of fenpyroximate and its Z-isomer (Z-fenpyroximate) in citrus fruits collected from 12 fields under good agricultural practices (GAPs). The average recoveries of fenpyroximate in whole fruits and citrus flesh were 104–110% and 92–109%, respectively, with corresponding RSDs of 1–4% and 1–3%. The average recoveries of Z-fenpyroximate were 104–113% and 90–91%, respectively, with RSDs of 1–2% in both cases. Each limit of quantification (LOQ) was 0.01 mg kg^−1^. Fifteen days after application with 56 mg kg^−1^, the terminal residues of fenpyroximate in whole fruits and citrus flesh were <0.010–0.18 mg kg^−1^ and <0.010–0.063 mg kg^−1^, respectively; the corresponding values for total fenpyroximate (the sum of fenpyroximate and Z-fenpyroximate) were <0.020–0.19 and <0.020–0.053 mg kg^−1^. The levels of terminal fenpyroximate residues in citrus fruit were less than the maximum residue limits (MRLs) specified in all the existing international standards. In addition, the risk quotients RQ_c_ and RQ_a_ were both less than 100%, indicating that the long-term and short-term dietary intake risks posed to Chinese consumers by fenpyroximate in citrus fruit are both acceptable after a 15-day harvest interval.

## 1. Introduction

Citrus belongs to the citrus subfamily of the rutaceae family. Citrus fruits are low in protein and fat but rich in nutrients such as vitamins, dietary fiber, flavonoids, minerals, and carotenoids; they provide good health benefits to the human body [1,2]. Globally, citrus is the most extensively grown category of fruit [3,4]; in China, citrus plants are widely cultivated in regions such as Hubei, Zhejiang, Sichuan, and Shanxi, accounting for about 30% of the world’s total citrus production, making China the world’s largest producer—and consumer—of citrus fruits [5,6]. Because citrus plants are perennial, they are vulnerable to diseases and pests during growth [7]. Any delays or failures in preventative measures can result in large-scale cross-infections, which may affect yield and quality, and cause slow growth or death in affected plants.

Fenpyroximate (E isomer; Figure 1) is the common name for tert-butyl (E)-alpha-(1,3)-dimethyl-5-phenoxy-1H-pyrazol-4-yl methyleneamino-oxy)-para-toluate [8], which was developed in 1985 by Nihon Noyaku Co. (Tokyo, Japan) [9]. Fenpyroximate has an oxime-bearing pyrazole structure, and is now widely used as an acaricide. It is characterized by its strong contact action upon harmful mites such as red spiders and whole-claw spiders, and it exhibits a high degree of fast-acting acaricidal activity [10]. The action mechanism of fenpyroximate is different from that of general insecticides; it mainly interferes with neurophysiological activities, and is inhibitory on nerve conduction; as a result, mites and insects experience the symptoms of paralysis, inactivity, cessation of food intake, and, ultimately, death, 2–4 days after coming into contact with fenpyroximate [11]. As a pesticide, fenpyroximate has the advantages of high efficiency, a broad spectrum, and a long effective period, with no cross-resistance to other pesticides. In China, fenpyroximate is currently registered on corn, citrus, apple, and lycium labels [12].

In previous studies, the methods of QuEChERS and micro-solid-phase extraction (DMSPE), as well as high-performance liquid chromatography (HPLC) and ultra-high-performance liquid chromatography–tandem mass spectrometry (UHPLC-MS/MS), have all been used to analyze the fenpyroximate in fruit juice, apple, tea, ginger, cucumber, green cabbage, orange, guava, eggplant, herbal medicines, and kiwi fruit [9,10,13,14,15]. However, these studies only detected residual levels of the primitive body of fenpyroximate. To date, no studies have sought to simultaneously detect fenpyroximate and its Z-isomer in citrus; more specifically, no studies have used multi-regional field trials to assess the dietary intake risks posed by fenpyroximate and its Z-isomer in citrus fruits.

In the 2013 evaluation report of the FAO/WHO Joint Meeting on Pesticide Residues (JMPR), fenpyroximate was defined as fenpyroximate (E-fenpyroximate) for the risk assessment [16]. However, for the 2017 JMPR report, fenpyroximate was re-evaluated, and a risk assessment was carried out for a combination of fenpyroximate (E-fenpyroximate) and its Z-isomer (M-1, Z-fenpyroximate). As a result, it is now a matter of importance to establish an analytical method to determine the residual behavior of fenpyroximate and its Z-isomer, and to assess any related dietary intake risks [8].

The objectives of the present study were as follows: (1) To establish a simple and accurate method of QuEChERS combined with UHPLC–MS/MS to measure fenpyroximate and its Z-isomer in citrus; (2) to study the terminal residues of fenpyroximate and its Z-isomer in citrus fruit grown in fields under GAPs; and (3) to carry out a dietary risk assessment based on the residue data obtained from supervised trials and the dietary consumption of Chinese residents. In summary, we sought in this work to evaluate the potential health risk posed to Chinese residents by fenpyroximate ingested from citrus, and also provide a proposal for the safe and rational use of this pesticide in citrus fruit cultivation.

## 2. Results and Discussion

### 2.1. Method Validation

A pesticide residue analysis includes sample pre-treatment and sample detection. In this experiment, we used a flow injection method to inject a mixed standard solution of fenpyroximate and Z-fenpyroximate into a mass spectrometer at 1 mg/L, and the mass spectrum was scanned with the positive ionization mode (ESI+) in the MRM mode. By means of primary mass spectrometry scanning, the value of the parent ion was determined to be 422.2, and the value of the declustering potential (DP) was optimized so that strongest signal occurred at 90 V. The values of the pieces were determined to be 366.1 and 135.0 using product ion scanning. The collision energy (CE) of the mass spectrometry was optimized to maximize the response of product ions at 23 V and 43 V, respectively. In order to separate the fenpyroximate and Z-fenpyroximate as much as possible, a slow flow rate of 0.2 mL min^−1^ was set. Using a single-pesticide standard solution for full scanning, the retention times of fenpyroximate and its Z-isomer were determined to be 2.71 and 2.28, respectively, under the mass spectrometry conditions (Table 1); then, a mixed standard solution of the two pesticides was scanned. As can be seen from the chromatogram shown in Figure 2, and the total ion chromatogram and MS spectra shown in Appendix A, respectively, the method produced a good separation effect. Acetonitrile is a highly polar solvent that exhibits good solubility with the majority of pesticides and strong permeability to sample matrixes; it can effectively remove impurities such as oils and pigments from samples [9]. In this study, acetonitrile was used as the extraction solvent for the components to be tested. In the present study, dispersive solid-phase extraction was used for purification; however, PSA (N-propyl ethylenediamine), C18, and graphitized carbon black (GCB) have often been used as purification agents for analyzing pesticide residues in agricultural products. Farag Malhat [9] described the purification of fenpyroximate in blank guava, orange, and eggplant samples using PSA and GCB. The satisfactory results reported were due to the high content of sugars, organic acids, and pigments in citrus. For the present study, PSA and GCB were selected as purification materials, and the recovery rate was good; the solution was close to colorless when the levels of PSA and GCB were 50 mg and 5 mg, respectively.

### 2.2. Method Validation

In order to analyze the experimental samples accurately, the main parameter indicators of the analysis methods for the two pesticides in whole fruits and citrus flesh were validated and evaluated; these parameters included linearity, matrix effect, accuracy, precision, and sensitivity.

#### 2.2.1. Linearity and Sensitivity

Fenpyroximate and its Z-isomer in two matrixes were verified using linearity, the correlation coefficient (R^2^), and the LOQ. As can be seen in Table 2, there was a good linear relationship between the peak areas and the concentrations of the two tested compounds in the range of 0.01–0.5 mg kg^−1^, and the linear correlation coefficients (R^2^) were greater than 0.99. The LOQ was identified as the minimum concentration for analyzing the actual samples that could be determined with acceptable precision and accuracy under the stated conditions; this was found to be 0.01 mg kg^−1^ for both compounds in samples of whole fruit and citrus flesh.

#### 2.2.2. Recovery

To evaluate the accuracy and precision of the detection method, a recovery test was used to calculate the average recovery and relative standard deviation (RSD) for each of the two tested compounds in both whole fruit and citrus flesh. The results are shown in Table 3. It can be seen that the average recoveries of fenpyroximate in the matrixes of whole fruit and citrus flesh were 104–110% and 92–109%, respectively, with RSDs of 1–4% and 1–3%. The average recoveries of Z-fenpyroximate were 104–113% and 90–91%, respectively, and all RSDs were in a range of 1–2%, thus meeting the requirements of accuracy and precision for the detection method of pesticide residues.

#### 2.2.3. Matrix Effect

Matrix effects (MEs) should be avoided as much as possible when using UHPLC–MS/MS for residue determination [17]. It can be seen from Table 2 that the whole fruit and citrus flesh exhibited different levels of a matrix inhibition effect on the two compounds. The ion suppression percentages of fenpyroximate and Z-fenpyroximate in whole citrus fruit were −16.8% and −25.8%, respectively; the corresponding ion suppression percentages in citrus flesh were −12.0% and −6.18%, respectively. Generally, it may be said that any inhibition or enhancement effect originates from co-extraction compounds such as phospholipids, sugars, phenols, and pigments. However, the underlying mechanisms of the matrix effects have not yet been fully explained [18]. Compared to citrus flesh, the matrix effect of whole citrus fruit was greater, possibly due to the presence of more co-extraction compounds. In order to eliminate the influence of matrix effects on the detection results, this experiment uniformly used a matrix-matching standard curve for the quantitative analysis of substances.

### 2.3. Terminal Residues

For the terminal residue experiment, fenpyroximate was applied once to citrus from all twelve fields at the registered high dosage (56 mg kg^−1^). The terminal residues of the samples were collected at 15 and 25 days after application, and the total residues of fenpyroximate were calculated. The results are shown in Table 4 and Appendix A, and the quality control (QC) during the actual sample test is shown in Appendix A.

In the samples of whole fruit and citrus flesh, the residual amounts of fenpyroximate were found to be <0.010–0.18 and <0.010–0.063 mg kg^−1^, respectively; the corresponding values for total residues were <0.020–0.19 and <0.020–0.053 mg kg^−1^; in the case of Z-fenpyroximate, all the values were less than 0.010 mg kg^−1^. The maximum residue limits (MRLs) of fenpyroximate in citrus fruits determined by the CAC [19], the United States [20], and Japan [21] are 0.6, 0.5, and 2 mg kg^−1^, respectively. In addition, China [22], the European Union [23], and South Korea [24] have established MRLs for fenpyroximate in citrus of 0.2, 0.5, and 0.5 mg kg^−1^, respectively. In this study, the terminal residues in citrus were found to be below all the above-mentioned limit values.

Throughout the experiment, in both whole fruit and citrus flesh, the residual concentrations of fenpyroximate, Z-fenpyroximate, as well as total residues all exhibited a decreasing trend with increases in the harvest interval. In a small number of cases, the residue concentrations after long intervals were slightly higher than the corresponding short-interval values; however, this may have been the result of uneven sampling, quartering, or scaling during sample preparation so that the experimental results were not affected. The residual concentrations of fenpyroximate in citrus flesh were lower than those in entire fruits because droplets of pesticides wholly adhered to the surface of the citrus peel after application. The total residues of fenpyroximate in whole fruit and citrus flesh were all derived from fenpyroximate because the chemical configuration of fenpyroximate in plants gives it an advantage compared to Z-fenpyroximate [25].

### 2.4. Long-Term Dietary Risk Assessment

The long-term dietary risk assessment was based on data from monitoring surveys on the nutritional and health status of Chinese residents; these were combined with STMR or MRL so that a national estimated daily intake (NEDI) could be determined. This was used to evaluate the dietary intake risk posed by edible parts of foods such as citrus flesh. At present, based on the data from the Chinese pesticide information network, the crops which are registered for fenpyroximate in China include maize, citrus, apple, and lycium [12]. In line with the principle of maximizing risk, we used MRLs from countries where crops are registered for fenpyroximate, with priority given to Chinese standards. As citrus and apple both belong to the “fruit” classification, a fenpyroximate STMR (total amount) of 0.020 mg kg^−1^ in citrus flesh collected at 15-day intervals was used in the present study to represent the fruit classification. The average weight of Chinese adults (bw) is 63 kg, and the allowable daily intake (ADI) of fenpyroximate is 0.01 mg kg^−1^ bw [8]. Table 5 shows the long-term dietary intake risk posed by fenpyroximate in citrus. On the basis of Chinese dietary data expressed on a per capita basis, the NEDI of fenpyroximate (total amount) for the general population was calculated to be 3.98 × 10^−4^ mg kg^−1^ bw. The value of RQ_c_ was 3.98%, which was considerably less than 100%, indicating that, after a 15-day harvest interval, the long-term dietary intake risk of fenpyroximate in citrus was very low, and acceptable for Chinese consumers.

### 2.5. Short-Term Dietary Risk Assessment

The short-term dietary risk posed by fenpyroximate after the intake of citrus was assessed for two different groups (children aged 1–6 years, and the general population aged >6 years). According to data from the World Health Organization [26], the large portion of dietary consumption (LP) for children in China is 0.587 kg; for the general population, the figure is 1.014 kg. The unit weight (U_e_) of a citrus is 0.124 kg, with a variation factor (υ) of three. The average weight of children aged 1–6 years in China is 16.1 kg; the average weight of the general population is 63 kg. In this experiment, the highest residue (HR) of fenpyroximate in citrus was found to be 0.053 mg kg^−1^. The acute reference dose (ARfD) is 0.01 mg kg^−1^ bw [8]. The short-term dietary risks posed by fenpyroximate in citrus to children aged 1–6 and the general population in China are shown in Table 6. Among the two groups of people, the national estimated short-term intake (NESTI) of fenpyroximate was 0.00275 mg kg^−1^ bw and 0.00106 mg kg^−1^ bw, respectively. The RQ_a_ values were 27.5% for children and 10.6% for the general population. Both these values were well below 100%, indicating an acceptable level of dietary intake risk posed by fenpyroximate in citrus, both to children aged 1–6 and to the general population in China.

## 3. Materials and Methods

### 3.1. Reagents and Chemicals

The fenpyroximate standard (Solid, purity 97.79%) and Z-fenpyroximate standard (liquid, 100.0 µg mL^−1^) were provided by Dr. Ehrenstorfer (Augsburg, Germany). The 28% suspension concentrate (SC) of fenpyroximate was supplied by Shanxi Huarongkaiwei Biotech Co., Ltd. (Xi’an, China). Formic acid and chromatographic-grade methanol were purchased from Merck AG (Darmstadt, Germany) and Thermo Fisher Scientific Co., Ltd. (Shanghai, China), respectively. Acetonitrile and HPLC-grade ammonium acetate were purchased from Tiandi Co., Ltd. (Dublin, OH, USA) and the Tianjin Guangfu Fine Chemical Research Institute (Tianjin, China), respectively. Analytical-grade sodium chloride was purchased from Sinopharm Chemical Reagent Co., Ltd. (Shanghai, China). Purification tubes of 2 mL capacity (150 mg anhydrous MgSO_4_, 50 mg PSA, and 5 mg GCB) were purchased from Shimadzu Jier Trading Co., Ltd. (Shanghai, China).

A standard stock solution of fenpyroximate was prepared by weighing 10.0 mg of the fenpyroximate standard and then dissolving it with acetonitrile in a 100 mL volumetric flask. The standard stock solutions of fenpyroximate and Z-fenpyroximate 100 µg mL^−1^ were all stored in a freezer at −18 °C.

We placed 0.1 mL of fenpyroximate and Z-fenpyroximate in a 10 mL volumetric flask with acetonitrile to prepare a mixed standard solution of 10 µg mL^−1^. The mixed standard solution was serially diluted with whole fruit and citrus flesh at concentrations of 0.01, 0.02, 0.05, 0.1, 0.2, and 0.5 µg mL^−1^. All solutions were stored in a dark place at 4 °C.

### 3.2. Field Trials

In line with the guideline requirements for the testing of pesticide residues in crops (NY/T 788-2018) [27], the terminal residue experiment was conducted between September 2022 and January 2023 at 12 locations in China: the city of Longnan in the Gansu province; the city of Hangzhou in the Zhejiang province; the Zhangjiajie and Changsha cities in the Hunan province; the city of Lichuan in the Hubei province; the city of Nanning in the Guangxi province; the district of Beibei in the city of Chongqing; the southwestern district in the Guizhou province; the city of Yuxi in the Yunnan province; the Guangzhou and Maoming cities in the Guangdong province; and the city of Danzhou in the Hainan province. The soil properties and climatic conditions of the field experiment are shown in Appendix A. Tests were carried out in areas with four trees, with isolation zones between the tested areas and a blank control area to avoid cross-contamination.

In order to measure the levels of terminal and dissipation residue of fenpyroximate and its Z-isomer, a dosage of 56 mg kg^−1^ fenpyroximate SC (a registered high dosage in citrus) was applied once to the citrus. Terminal residue samples of the whole fruit and citrus flesh were collected 15 d and 25 d after application. Two independent samples were collected, each involving at least 12 fruits randomly chosen from different places on the tree. Samples were cut into pieces of less than 1 cm^2^ and then mixed and divided into pieces of 150 g weight by means of quartering. All samples were stored at −20 °C in preparation for testing.

### 3.3. Sample Pre-Treatment

A QuEChERS-based method [28] was used for the pre-treatment of samples in this experiment. Uniformly crushed samples were weighed (10.00 g of both whole fruit and citrus flesh) and placed into a 50 mL centrifuge tube, to which 10 mL acetonitrile and 5.0 g sodium chloride were added. Extraction was carried out for 10 min at 2500 rpm using vortex oscillation, followed by centrifuging at 8000 r min^−1^ for 3 min. A 1.5 mL amount of supernatant 1.5 mL was sucked out and transferred to a 2 mL centrifuge tube filled with 150 mg anhydrous MgSO_4_, 50 mg PSA, and 5 mg GCB. The tubes were vortexed at 2500 rpm for 5 min and centrifuged at 5000 rpm for 2 min. Finally, the purified liquid was filtered through a 0.22 µm membrane filter for analysis via UHPLC-MS/MS.

### 3.4. Instrumental Parameters

Chromatographic conditions: UHPLC–MS/MS analyses of fenpyroximate and its Z-isomer were carried out using SCIEX Triple Quad 4500 (AB company, Framingham, MA, USA). The chromatographic column was ACQUITY UPLC^®^ BEH 1.7 µm C18 100 A (100 × 2.1 mm, SN: 02973711615165). The temperatures of the column and sample plate were 30 °C and 20 °C, respectively. The mobile phase was composed of 90% aqueous phase (4 mmol L^−1^ ammonium acetate aqueous solution with 0.1% formic acid) and 10% organic phase (pure methanol), flowing at 0.2 mL min^−1^. The injection volume was 2 µL.

Mass spectrum conditions: The mass spectrum was scanned at positive ionization mode (ESI+) in MRM mode. The ionization voltage was 5500 V, the ion source temperature was 550 °C, and the collision gas was nitrogen. The other mass-spectrometry parameters for fenpyroximate and its Z-isomer are shown in Table 1.

### 3.5. Method Validation

The analytical methods used for fenpyroximate and Z-fenpyroximate in whole fruit and citrus flesh were verified with respect to linearity, matrix effect (ME), accuracy, precision, and sensitivity, in conformity with SANTE/11312/2021 [29]. The mixed standard solutions were diluted with blanks of whole citrus fruit, blanks of flesh citrus, and acetonitrile, to prepare a series of standard working solutions with concentrations of 0.01, 0.02, 0.05, 0.1, 0.2, and 0.5 μg mL^−1^. A calibration curve was then drawn with the mass concentration as the X-axis and the corresponding peak area as the Y-axis; the linear correlation coefficient R^2^ was expected to be more than 0.99.

Three levels of mixed standard solutions (0.01, 0.1, and 0.2 mg kg^−1^) were added to blank matrixes of whole fruit and citrus flesh, so that the recovery and relative standard deviation (RSD) values could be calculated for the target compounds, enabling the accuracy and precision of the method to be evaluated. In order to cover the maximum detected concentration among the testing samples, we added a concentration of 0.5 mg kg^−1^ to the whole fruit matrix. Five repeated tests were carried out for each level. The recovery rate of the method was found to be 70–120%, and the RSD was less than 20%, which was considered a satisfactory degree of accuracy. Usually, the lowest spiked level was set as the limit of quantification (LOQ) to evaluate the sensitivity of the method.

As reported in the literature [30,31], the ion suppression percentage is commonly used to evaluate matrix effects with the following calculation formula:Ion suppression percentage = (K_m_ − K_s_)/K_s_(1)
where K_m_ and K_s_ are the slopes of the matrix-matched standard curve and the solvent standard curve, respectively. An ion suppression percentage of 0 indicates no matrix effect; an ion suppression percentage of more than 0 indicates that the sample matrix has a matrix-enhancing effect on the determination of the target compound; and ion suppression percentages of less than 0 indicates a matrix-inhibiting effect on the determination of the target compound.

### 3.6. Definitions of Fenpyroximate

For plant commodities, in line with the JMPR research results, the residue was defined as fenpyroximate for compliance with MRLs, and for the dietary risk assessment as the sum of the parent fenpyroximate and its Z-isomer (M-1), expressed as fenpyroximate and calculated directly using addition [8].

### 3.7. Assessment of Long-Term Dietary Intake Risk

The method for assessing long-term dietary intake risk was recommended by the review committee of the National Pesticide Residue Standards. The risk quotient (RQ_c_) was used to represent the long-term dietary intake risk, and the calculation formulas were as follows [32]:NEDI = (∑STMR_i_ × F_i_)/bw(2)
RQ_c_ = NEDI/ADI × 100%(3)
where NEDI (mg kg^−1^ bw) refers to the national estimated daily intake; STMR_i_ (mg kg^−1^) is the median residue of total fenpyroximate (the sum of fenpyroximate and Z-fenpyroximate) in citrus flesh collected from field trials; F_i_ (kg d^−1^) is the per capita daily consumption of citrus in China; bw (kg) is the average weight of Chinese adults; ADI (mg kg^−1^ bw) is the allowable daily intake of fenpyroximate taken as determined by the Joint FAO/WHO Meeting of Pesticide Residues (JMPR) [8]. A result of RQ_c_ ≤ 100% indicates that the level of risk is acceptable to consumers, whereas RQ_c_ > 100% indicates a level which is unacceptable.

### 3.8. Assessment of Short-Term Dietary-Intake Risk

Assessment of short-term dietary intake risk was calculated in line with case 2a of the JMPR-recommended method using the following formulas [32]:NESTI = [U_e_ × HR × υ + (LP − U_e_) × HR]/bw(4)
RQ_a_ = NESTI/ArfD × 100%(5)
where NESTI (mg kg^−1^ bw) refers to the national estimated short-term intake; HR (mg kg^−1^) is the highest total of fenpyroximate residue in citrus flesh collected from field trials; U_e_ (kg) is the weight of a single orange; υ is a variation factor; LP (kg) represents a large portion of dietary consumption; Bw (kg) is the average weight of the consumer population; and ARfD (mg kg^−1^ bw) is the acute reference dose taken from JMPR [8,12]. The short-term dietary-intake risk value is expressed as RQ_a_. When RQ_a_ ≤ 100%, short-term dietary risk is at an acceptable level; when RQ_a_ > 100%, the level of risk is not acceptable.

## 4. Conclusions

In this study, a qualitative and quantitative method of analysis was established to detect fenpyroximate and its Z-isomer in whole-fruit and flesh citrus. This method was found to be simple, time-saving, effective in purification, sensitive, and accurate. This method is characterized by an excellent linear relationship, with levels of accuracy and precision which meet the requirements for detection and analysis of pesticide residues. Using this method, we obtained a short-term separation of fenpyroximate and its Z-isomer, and the effect of the separation was good. The terminal residues of fenpyroximate in whole-fruit and flesh citrus collected from twelve fields were <0.010–0.18 and <0.010–0.063 mg kg^−1^, respectively, which were lower than the Chinese MRL values. The terminal residues of fenpyroximate (the sum of fenpyroximate and Z-fenpyroximate) in whole-fruit and flesh citrus were <0.020–0.19 and <0.020–0.053 mg kg^−1^, respectively, and RQ_c_ and RQ_a_ were 3.98% and 10.6–27.5%, respectively, i.e., less than 100%, indicating that the levels of long-term and short-term dietary-intake risk posed to Chinese consumers by fenpyroximate in citrus were acceptable, given a 15-day harvest interval after application at 56 mg kg^−1^. In conclusion, this study provides a scientific basis for the safe use of fenpyroximate in citrus as well as basic data for revising the MRL values.

## Figures and Tables

**Figure 1 molecules-28-07123-f001:**
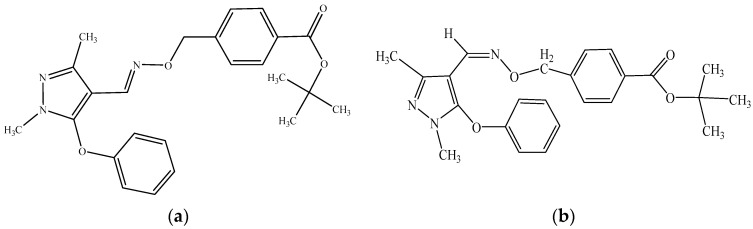
Structural formulas of (**a**) fenpyroximate and (**b**) Z-fenpyroximate.

**Figure 2 molecules-28-07123-f002:**
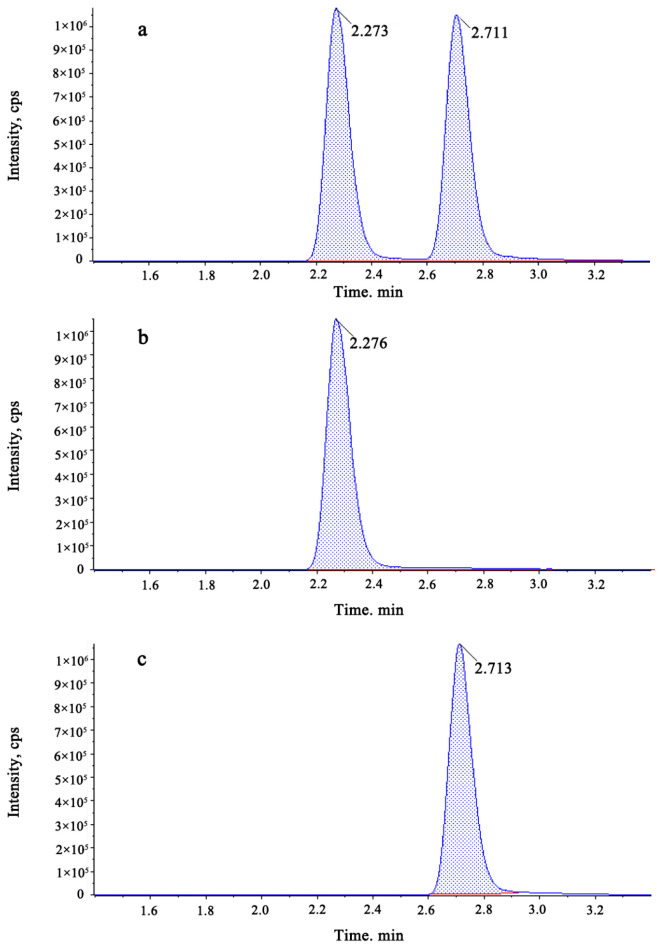
Chromatograms of fenpyroximate and Z-fenpyroximate: (**a**) retention time for simultaneous detection of fenpyroximate and Z-fenpyroximate and (**b**,**c**) retention times for single detections of Z-fenpyroximate and fenpyroximate, respectively.

**Table 1 molecules-28-07123-t001:** Retention times and UHPLC–MS/MS mass spectral parameters for fenpyroximate and Z-fenpyroximate.

Compound	Retention Time (R_t_, min)	Qualitative Ion Pair (M Z^−1^)	Quantitative Ion Pair (M Z^−1^)	Declustering Potential (DP, V)	Collision Energy (CE, V)
Fenpyroximate	2.71	422.2 > 366.1	422.2 > 135.0	90	23
Z-fenpyroximate	2.28			90	43

**Table 2 molecules-28-07123-t002:** Calibration curves, correlation coefficients (R^2^), MEs, and LOQs of fenpyroximate and Z-fenpyroximate in matrixes.

Compounds	Matrix	Calibration Curve	R^2^	ME/%	LOQ/ (mg kg^−1^)
Fenpyroximate	Whole citrus	y = 43,651,900x + 85,383	0.9916	−16.8	0.01
	Citrus flesh	y = 46,191,900x + 103,438	0.9913	−12.0	0.01
	Acetonitrile	y = 52,490,800x + 117,794	0.9904	-	0.01
Z-fenpyroximate	Whole citrus	y = 38,951,300x + 40,262	0.9920	−25.8	0.01
	Citrus flesh	y = 49,233,100x + 104,127	0.9913	−6.18	0.01
	Acetonitrile	y = 52,475,700x + 147,634	0.9907	-	0.01

**Table 3 molecules-28-07123-t003:** Average recoveries (*n* = 5) and RSDs of fenpyroximate and Z-fenpyroximate in matrixes.

Compounds	Matrix	Spiked Level/ (mg kg^−1^)	Average Recovery (*n* = 5)/%	RSD/%
Fenpyroximate	Whole citrus	0.01	109	2
		0.1	110	1
		0.2	108	1
		0.5	104	4
	Citrus flesh	0.01	109	2
		0.1	98	3
		0.2	92	1
Z-fenpyroximate	Whole citrus	0.01	113	2
		0.1	110	2
		0.2	104	1
		0.5	108	1
	Citrus flesh	0.01	91	2
		0.1	90	1
		0.2	90	2

**Table 4 molecules-28-07123-t004:** Final residues, STMRs, and HRs of each compound in different matrixes.

Application Dose (mg kg^−1^)	Application Times	Harvest Interval (Days)	Matrix	Residues (mg/kg)				
				Fenpyroximate	Z- Fenpyroximate	Total Residues (Evaluate Definition)	STMR (mg kg^−1^)	HR (mg kg^−1^)
56	1	15, 25	Whole citrus	<0.010–0.18	<0.010	<0.020–0.19	-	-
			Flesh citrus	<0.010–0.063	<0.010	<0.020–0.053	0.020	0.053

**Table 5 molecules-28-07123-t005:** The long-term dietary intake risk posed by fenpyroximate in citrus under Chinese dietary conditions.

	Food Classification	F_i_ (kg)	Residue (mg kg^−1^)	Sources	NEDI (mg kg^−1^ bw)	Allowable Daily Intake (mg kg^−1^ bw)	RQ_c_/%
Total fenpyroximate (sum of fenpyroximate and Z-fenpyroximate)	Other grains	0.0233	0.01	CAC, MRL (maize)	3.70 × 10^−6^	ADI	NEDI/ADI × 100%
	Fruits	0.0457	0.020	STMR in this study (flesh citrus)	1.45 × 10^−5^		
	Salt	0.012	2	China, MRL (lycium)	3.80 × 10^−4^		
	total				3.98 × 10^−4^	0.01	3.98

**Table 6 molecules-28-07123-t006:** The short-term dietary intake risk posed by fenpyroximate in citrus to two age groups within the Chinese population.

Age	bw (kg)	LP (kg)	NESTI (mg kg^−1^ bw)	RQ_a_ (%)
1–6 years	16.1	0.587	0.00275	27.5
>6 years	63	1.014	0.00106	10.6

## Data Availability

Data presented in this research are available from the corresponding author for request.

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
