# Peer review of "Measuring the Residual Levels of Fenpyroximate and Its Z-Isomer in Citrus Using Ultra-High-Performance Liquid Chromatography–Tandem Mass Spectrometry and Assessing the Related Dietary Intake Risks"

_molecules, 2023, doi:10.3390/molecules28207123_

Round 1
Reviewer 1 Report
The publication is interesting and may be accepted to Molecules, but the authors need to answer some questions
Was the LC-MS method optimized? How were the values of MS operating parameters selected? How were LC conditions chosen that two compounds with identical masses separated using a simple C18 column?
What was the purification of the sample using QuEChERS? The matrix effect indicates that incomplete, was that the case? What compounds remained in the extract after extraction? Please show MS spectra, TIC chromatograms in the supplement.
For what reasons was QuEChERS chosen? Were other techniques considered, or additional purification of the sample after QuEChERS?
Author Response
Thank you very much for expert opinions, which has greatly improved my article. Based on the opinions, I have answered the questions and made the following modifications:
1. MS method has been optimized as follow: The experiment used a flow injection method to inject a mixed standard solution of fenpyroximate and Z-fenpyroximate for 1 mg/L into the mass spectrometer, and the mass spectrum was scanned at positive ionization mode (ESI+) in MRM mode. The parent ion was determined to be 422.2 through primary mass spectrometry scanning, and the value of declustering potential (DP) was optimized to have the strongest signal at 90 V. The pieces were determined to be 366.1 and 135.0 through product ion scanning. The collision energy (CE) of the mass spectrometry was optimized to maximize the response of product ions at 23 V and 43 V, respectively. Above-mentioned contents have been added to lines 78-86 of the paper.
Sample pre-treatment refers to reference [9] ‘Reside of fenpyroximate acetate in/on guava, orange, and eggplant under open field conditions’. The extraction agent was acetonitrile, and the purification agent were PSA and GCB. The selection of LC conditions depends on our research team's many years of experience in pesticide residue research for agricultural products. Considering that fenpyroximate is a medium polarity compound, and referring to relevant references, it is suitable to separate such compounds using C18 columns. In order to separate two compounds with the same quality as much as possible, a column with the length of 100 mm was selected and the mobile phase was slowly enough with the flow rate of 0.2 mL min-1 was setted, Figure 2 showed that the separation effect was very good.
LC and MS conditions are derived from paper 3.4 The Instrument Parameters section.
2. The QuEChERS method is a pre-treatment method for pesticide residue samples. In this experiment, the organic phase was mainly extracted by rapidly vibration of acetonitrile, and then purified by PSA and GCB as purification materials, PSA was mainly used to adsorb sugar and organic acids in citrus, GCB was mainly used to adsorb pigments in citrus, and finally the extracting solution was filtered through a 0.22 µm membrane filter. Paper of 3 The Sample pre-treatment provided a detailed description of the pre-treatment steps for samples using QuEChERS.
Combining the calculation formula for matrix effects in lines 292-299 and the calculation results in lines 138-142, the result indicated that it has matrix effects, both the whole fruit and flesh of citrus exhibited matrix inhibitory effect.
Reference [9] mentioned that citrus have large amount of sugar, organic acids, and fragments.
MS spectra and TIC chromatograms were shown as Figure 1S and Figure 2S in the supplement.
3. QuEChERS is a sample pre-treatment method developed by Anastasiades in 2002. ‘QuEChERS’ comes from the abbreviation of quick, easy, check, effective, rugged, and safe, which is a ‘fast, simple, cheap, effective, stable, and safe’ extraction method. Due to its advantages of simple operation, low cost, and short analysis time, it has been widely used in pesticide residues analysis in recent years (References [2][7][9][13]). This article does not considered other techniques or other purification after QuEChERS.
Reviewer 2 Report
This paper evaluated the potential health risk of fenpyroximate ingested from citrus to Chinese residents, and also provide proposal for the safe and rational use of this pesticide in citrus.
However, there are still the following issues:
1. The written expression is not standardized enough. Please revise it carefully to avoid confusion in Chinese and English words order, and pay attention to brevity and clarity.
2. The measurement unit of mg/kg should be unified as mg kg-1. Table 6 is not linked to the original text, etc. Please modify the figures and tables format according to the template carefully.
3. The summary mentions that “the terminal residuals of fenpyroximate in the city were less than the maximum residue limits (MRLs) setting for China”, The text 2.3. Terminal residues mentions that the limit value is lower than six countries or regions, including CAC and United States, should be uniformly stated as “lower than the existing international standard of MRLs”.
4. “Table 4 The Final residues of fenpyroximate, Z-fenpyroximate, and total residues in citrus” should be listed in the Supplementary Materials, where the final residues ranges, STMR and HR of each compound in different matrices should be listed here.
The written expression is not standardized enough. Please revise it carefully to avoid confusion in Chinese and English words order, and pay attention to brevity and clarity.
Author Response
Thank you very much for expert opinions, which have greatly improved my article. Based on the opinions, I have made the following modifications:
1. The author has carefully reviewed the paper with highlight.
2. Changed the measurement unit of mg/kg to mg kg-1 in line 22 of this article. Table 6 has already linked to the original text of line The formats of Table S1 and Table S2 have been modified based on the template.
3. Line 22 of the summary ‘The terminal residues of fenpyroximate in citrus was less than the maximum residue limits (MRLs) setting for china’has been changed to ‘The terminal residues of fenpyroximate in citrus was less than the maximum residue limits (MRLs) setting for all the existing international standards’.
4. The original Table 4 ‘The Final residues of fenpyroximate, Z-fenpyroximate, and total residues in citrus’has been replaced and listed in the supplementary materials, the new Table 4 contains final residues ranges, STMR and HR of each compound in different Matrixes.